# Multiplex Reverse Transcription Polymerase Chain Reaction Combined with a Microwell Hybridization Assay Screening for Arbovirus and Parasitic Infections in Febrile Patients Living in Endemic Regions of Colombia

**DOI:** 10.3390/tropicalmed8100466

**Published:** 2023-10-06

**Authors:** Paula Calderon-Ruiz, Gregor Haist, Annina Mascus, Andres F. Holguin-Rocha, Philip Koliopoulos, Tim Daniel, Gabriel Velez, Berlin Londono-Renteria, Britta Gröndahl, Alberto Tobon-Castano, Stephan Gehring

**Affiliations:** 1Center of Pediatric and Adolescent Medicine, University Medical Center, 55131 Mainz, Germany; gregor.haist@web.de (G.H.); amascus@students.uni-mainz.de (A.M.); philip.koliopoulos@unimedizin-mainz.de (P.K.); timwilli.daniel@unimedizin-mainz.de (T.D.); groendahl@uni-mainz.de (B.G.); stephan.gehring@unimedizin-mainz.de (S.G.); 2Malaria Group, Lab 610, Faculty of Medicine, University of Antioquia, Medellin 050010, Colombia; gabrielj.velez@udea.edu.co (G.V.); alberto.tobon1@udea.edu.co (A.T.-C.); 3Department of Entomology, College of Agriculture, Kansas State University, Manhattan, KS 66506, USA; aholguin@ksu.edu; 4School of Public Health and Tropical Medicine, Tulane University, New Orleans, LA 70112, USA; blondono@tulane.edu

**Keywords:** acute febrile syndrome, Colombia, malaria, dengue, rapid tests, multiplex RT-PCR-ELISA

## Abstract

Acute febrile syndrome is a frequent reason for medical consultations in tropical and subtropical countries where the cause could have an infectious origin. Malaria and dengue are the primary etiologies in Colombia. As such, constant epidemiological surveillance and new diagnostic tools are required to identify the causative agents. A descriptive cross-sectional study was conducted to evaluate the circulation and differential diagnosis of six pathogens in two regions of Colombia. The results obtained via multiplex reverse transcription polymerase chain reaction combined with a microwell hybridization assay (m-RT-PCR-ELISA) were comparable to those obtained using rapid tests conducted at the time of patient enrollment. Of 155 patients evaluated, 25 (16.1%) and 16 (10.3%) were positive for malaria and dengue, respectively; no samples were positive for any of the other infectious agents tested. In most cases, m-RT-PCR-ELISA confirmed the results previously obtained through rapid testing.

## 1. Introduction

Acute febrile syndrome (AFS) is defined as a morbid state with sudden onset of fever that evolves in less than seven days in patients between 5 and 65 years of age in whom no specific signs or symptoms of infection are identified [1]. AFS is a recurring reason for medical consultations in Colombia, and a diagnostic challenge given the diversity of diseases that can contribute to its cause [2].

Epidemiological surveillance and diagnostic searches have focused upon a few causative agents, e.g., malaria, for which diagnosis is more readily accessible. It is estimated that in some malaria-endemic regions, 90% of patients with AFS receive antimalarial treatment based upon a presumptive diagnosis, although only 20 to 40% are confirmed cases [3]. As such, 60 to 80% of cases remain undiagnosed, underestimating the number of cases of acute fever that are due to other causes [4,5].

In Choco and the Uraba region of Antioquia, conditions are favorable for the transmission of viral infectious agents that cause both vector-borne and non-vector-borne febrile illnesses. Like other regions of the country, the availability of diagnostic tests for etiologic agents other than malaria is limited [1].

Most infectious diseases that cause AFS are associated with socioeconomic factors such as inequality and poverty in areas far away from population centers. These factors hinder access to healthcare services, timely diagnosis, and adequate treatment [6]. In addition, diagnostic difficulties, a shortage of resources, and the lack of protocols increase the likelihood of complications [7].

There are regions in Colombia that are simultaneously endemic for diseases like malaria and circulating arboviruses, e.g., dengue (DENV), chikungunya, and Zika. The latter exhibit endemic–epidemic patterns, as well as ecological and epidemiological characteristics similar to other circulating arboviruses that have been described in America, but not in the national territory. The prevalence of these infections has not been studied to any degree using precise diagnostic methods, e.g., molecular biology [8,9].

The symptomatology of AFS is usually non-specific, impeding clinical identification of the causative agent. The most common symptoms in addition to fever are myalgia, chills, asthenia, and arthralgia; symptoms such as headache and abdominal pain, however, can also present at the time of anamnesis [9]. The similarities in clinical signs and symptoms reveal the complexity of differentiating the pathologies, making laboratory confirmation and differential diagnosis essential [2]. AFS diagnosis requires a comprehensive approach to each patient; the wide-ranging causes require detailed clinical methodologies and the diagnostic tests to support them. Diagnosis and treatment are adversely affected in regions where access to specialized tests is not immediately available [10,11].

According to the Columbian Clinical Practice Guidelines, Diagnosis and Treatment of Malaria, published in 2022, malaria should be diagnosed via microscopy (thick drop and/or peripheral blood smear) or using rapid diagnostic tests. Molecular tests are not indicated for the clinical diagnosis of malaria; however, they are recommended for the confirmation of cases (mainly mixed malaria) where microscopy can underestimate and the sensitivity of rapid tests can be low (~75%) [12].

In the case of dengue, rapid tests are only indicated for clinical diagnosis and should be performed within the first five days after symptom onset. The only tests accepted as confirmatory, however, are DENV-specific IgM antibody tests performed after the fifth day, or virus-specific IgG tests performed 15 days after the onset of symptoms [13]. Virus-specific IgG tests can also be performed five days after a secondary infection. Molecular tests, i.e., reverse transcription polymerase chain reaction (RT-PCR) or viral isolation, must be performed in reference laboratories, which are difficult to access in most regions.

RT-PCR molecular tests are recommended for the clinical diagnosis of Zika, chikungunya, yellow fever, and West Nile viruses [14,15,16]. Serologic tests, which are available in reference laboratories, are also recommended for cases of chikungunya and yellow fever. This implies that differential diagnosis of these diseases is adversely affected by any delay that inhibits the rapid, symptomatic treatment of patients in remote areas.

In recent years, diagnostic methods that allow the simultaneous detection of several pathogens have shown great potential, especially in tropical and subtropical areas where co-infections are common [6]. The multiplex reverse transcription polymerase chain reaction combined with a microwell hybridization assay (m-RT-PCR-ELISA) is a well-established, low-cost technique that allows the simultaneous detection of viruses, parasites, and bacteria. It has already been validated for the clinical diagnosis of 19 respiratory viruses and was found useful in epidemiological surveillance studies that identified 9 mosquito-borne pathogens at the Center of Pediatric and Adolescent Medicine, University Medical Center in Mainz, Germany [17,18].

In Choco, 23,544 cases of malaria and 888 cases of dengue were reported in 2022; no cases of Zika, chikungunya, yellow fever, or West Nile virus infection were reported. In Antioquia, 7180 cases of malaria, 2229 cases of dengue, 3 cases of Zika virus disease, and 6 cases of chikungunya, but no cases of yellow fever or West Nile virus, were reported [19].

Tropical diseases represent approximately 17% of infectious diseases worldwide. Conducting epidemiological surveillance studies and continuing the search for sensitive and specific diagnostic methods that are easy to implement, low-cost, and reliable should continue to be a priority. This is true particularly in regions where access to healthcare services is difficult and conditions exist that increase the risk of contracting these diseases [20]. The aim of the current study was to identify *Plasmodium* and arbovirus infections using m-RT-PCR-ELISA in cases of AFS, which occurred in two regions of Colombia that are difficult to access diagnostically.

## 2. Materials and Methods

### 2.1. Participants

A descriptive cross-sectional study was conducted between October and December 2019 and April and May 2022 with patients selected at random from Hospital Ismael Roldan Valencia (Quibdo–Choco) and Hospitals Antonio Roldan Betancur and Francisco Valderrama Hospital (Turbo and Apartado–Antioquia) (Figure 1). The study included 155 patients older than 2 years of age who reported fever within 72 h prior to enrollment; a temperature equal or greater than 37.5 °C was considered fever. All patients or parents/guardians (in case of minors) signed informed consent at the time of admission. A questionnaire concerning sociodemographic data, signs, and symptoms was also completed.

### 2.2. Sample Collection and Procedures

Blood was collected from each patient, and SD BIOLINE malaria Ag P.f/P.v (Abbott, Chicago, IL, USA) and Panbio^®^ Dengue Early ELISA (Abbott, Lake Forest, CA, USA) rapid tests were performed. The HemoCue^®^ Hb 201 + System (Hemocue, Ängelholm, Sweden) was used to quantify hemoglobin. Drops of whole blood obtained from a tube without anticoagulant were immediately transferred to cuvettes used specifically for this purpose.

Subsequently, the tubes were centrifuged, and the serum or plasma was pipetted into a tube with Invitrogen RNA later™ Solution (Thermo Fisher Scientific, Waltham, MA, USA). The samples, stored initially at −20 °C, were transported in liquid nitrogen to the Malaria Working Group laboratory of University of Antioquia and stored at −80 °C. The samples were subsequently shipped and processed at the Center of Pediatric and Adolescent Medicine, University Medical Center, Mainz, Germany.

### 2.3. Multiplex RT-PCR-ELISA

The nucleic acids were extracted from samples using the Roche high-purity viral nucleic acid kit and the protocol provided by the company (Roche Diagnostics GmbH, Mannheim, Germany). Six pathogens, *Plasmodium*, DENV, Zika virus, chikungunya virus, yellow fever virus, and West Nile virus, were evaluated.

Reverse transcription (RT), based upon the protocol established by Puppe et al., was performed prior to multiplex PCR [18]. Briefly, the extracted nucleic acid was diluted in buffer containing dNTP (Roche Diagnostics GmbH, Mannheim, Germany), first strand buffer (Invitrogen, Carlsbad, CA, USA), DTT (Invitrogen), hexanucleotide mix (Roche Diagnostics GmbH), RNAsin (Promega, Madison, WI, USA), and SuperScript II (Invitrogen). RT was performed through incubation for 10 min at 25 °C, followed by 50 min at 42 °C, then 5 min at 90 °C.

For PCR, an aliquot of the RT reaction product was diluted in a solution of nuclease-free water, PCR buffer I (AccuPrime System, Invitrogen), Primer Mix A/B (Invitrogen), AccuPrime Taq DNA Polymerase (Invitrogen), and digoxigenin-11-dUTP (Roche Diagnostics). The protocol for PCR was 35 cycles at 94 °C for 30 s, 55 °C for 20 s, and 72 °C for 40 s. The primers and probes used were based upon the protocol previously published by us [17]. After multiplex PCR, the amplicons were separated according to size via agarose gel electrophoresis.

For ELISA, the PCR product was denatured and an aliquot was transferred to a well in a 96-welled plate that contained 3′-biotinylated capture antibody specific for the amplified target sequence [17]. Subsequently, the plates were incubated for >1.5 h at 37 °C. The plates were washed, anti-digoxigenin-peroxidase (Roche Diagnostics) was added, and the plates were re-incubated for 45 min and then rewashed. Finally, the ABTS substrate solution (Roche Diagnostics) was added to each well, and the plates were incubated for approximately 10 min at 37 °C. Values >0.4 OD were considered positive; values <0.2 OD were considered negative. A follow-up test was performed for values between 0.2 and 0.4.

### 2.4. Single-Primer RT-PCR

Samples positive for either dengue or malaria were analyzed via single-primer RT-PCR to determine the DENV serotype and *Plasmodium* species using protocols published previously [18,21]. Additional information regarding the specific primer sequences used and the determination of cutoff values were reported previously by us [17].

### 2.5. Statistical Analysis

A descriptive analysis of sociodemographic variables, signs and symptoms, and the results obtained by molecular and ELISA testing was performed. Medians and interquartile ranges were estimated for quantitative variables without normal distribution. Relative and absolute frequencies were estimated for qualitative variables. All statistical procedures were performed using R Statistical Software (v4.0.2; R Core Team 2020; The R Foundation for Statistical Computing, Vienna, Austria).

## 3. Results

One hundred and fifty-five patients with a history of fever during the preceding 72 h were enrolled in the study (Table 1). Of these patients, 81 were from Antioquia (Turbo, Apartado) and 74 were from Choco (Quibdo). The median age was 24 years, 83 were men, and 21.3% of enrollees reported a history of malaria. Most patients resided in urban areas.

The median vital signs reported here were within normal values, except for the heart rates of dengue- and malaria-positive patients, in whom a significant increase in the median and interquartile ranges of the data was observed (Table 2). The most common symptoms were headache, chills, sweating, and arthralgia.

Twenty-five patients were found positive for malaria by the rapid test conducted on the day of inclusion (Table 3). Nine and seven cases were reported for *P. vivax* and *P. falciparum* in Antioquia, respectively. M-RT-PCR-ELISA confirmed twenty-three (92%) of these positive cases. It was not possible, however, to determine the species for one of the samples that tested *Plasmodium*-positive. Five and four of the cases in Chocó were positive for *P. falciparum* and *P. vivax*, respectively. All of the cases reported as positive through rapid testing in Chocó were confirmed positive through m-RT-PCR-ELISA, identifying the same species in both tests.

Sixteen patients were found positive for DENV by the rapid test conducted on the day of enrollment: 10 cases were reported in Antioquia and 6 were reported in Chocó (Table 4). One patient in Antioquia, who tested positive on the day of enrollment, later tested negative through m-RT-PCR-ELISA. A second patient in Antioquia, however, initially reported negative according to the results of the rapid test, but subsequently tested positive through m-RT-PCR-ELISA. All patients in Chocó found positive via rapid test were confirmed positive either through m-RT-PCR-ELISA or single-primer PCR. Co-infections by serotypes DENV1 and DENV4 were the most frequent at both study sites. Serotypes DENV2 and DENV3 were detected in Antioquia, while serotype DENV1 was found in Chocó. No patient was found positive for Zika, chikungunya, yellow fever, or West Nile virus.

## 4. Discussion

Diagnosis of the pathogenic cause of AFS is based mainly upon anamnesis, physical examination, and laboratory confirmation when possible. An epidemiological characterization is often not sufficient to guide the diagnosis [9,22]. Although many pathogens that cause AFS can be treated therapeutically, not having a confirmatory diagnosis represents a challenge that can lead to major complications such as severe cases of leptospirosis, malaria, and dengue [22,23,24,25]. Malaria is the most prevalent disease in tropical and subtropical countries and should be considered first in cases of AFS that occur in places such as Colombia, where malaria is endemic [9]. However, arboviruses, e.g., DENV, have spread worldwide and pose an additional threat in many countries triggering health, social, and economic crises [6,21]. Despite the fact that malaria and dengue are the most common causes of AFS in endemic areas, the proinflammatory symptomatology reported here is similar to that found in cases of AFS in which no clinical etiology can be identified [1,6]. Therefore, the performance of diagnostic tests is essential for accurate diagnoses.

Currently, the methods used most often to diagnose AFS are based upon microscopic examination, rapid tests, molecular tests such as PCR, and serology [6]. The availability of diagnostic tools to resolve unidentified cases is limited [1]. The m-RT-PCR-ELISA panel used in the present study includes nine pathogens that can cause non-specific febrile illnesses. The methodology is based upon one that was first developed to identify the etiologic agents of acute respiratory infections [26]. Since its first introduction into routine clinical practice in 1996 at the Mainz University Medical Center, more than 30,000 patient samples have been analyzed through respiratory m-RT-PCR-ELISA [17]. The panel described in this article has been expanded since 2016 to include DENV, chikungunya, West Nile, yellow fever, Zika, Rift Valley fever, O’nyong-nyong, and Semliki-Forest viruses [27,28,29]. It is currently being used in epidemiological studies but has not been validated for clinical practice [27,28,29].

M-RT-PCR-ELISA panels are increasingly available commercially to accelerate and simplify the diagnosis of a variety of diseases in standard clinical practice [30,31]. The m-RT-PCR-ELISA test used in this project was developed to pursue the same goals. The 155 patient samples analyzed in this study confirmed the ability of m-RT-PCR-ELISA to detect the causative agents of disease.

The present study demonstrated the presence of both *P. vivax* and *P. falciparum* species, as well as four DENV serotypes (DENV1, DENV2, DENV3, and DENV4) in Antioquia, and two DENV serotypes (DENV1 and DENV4) in Chocó. These results are consistent with previous reports of the presence of four DENV serotypes in Colombia [32,33]. The results obtained in the current study revealed a higher frequency of patients co-infected with more than one serotype than patients infected with a single serotype only. These data differ from studies conducted in Brazil, Peru, and other hyperendemic regions of the world (e.g., Indonesia and Cameroon) that reported a lower frequency of concurrent DENV serotype infections [34,35,36,37].

The concurrent circulation of DENV serotypes is a common phenomenon in Latin America [38]. Whether coinfections influence the clinical outcome of patients is unresolved at this time; different studies have obtained different conclusions [39,40]. Most epidemiological and vaccine-related studies suggest, however, that a higher risk of antibody-dependent enhancement leading to severe dengue exists in places where several DENV serotypes co-circulate [41,42]. Additional studies involving a large number of cases are required to resolve the relationship between coinfections and disease severity.

The results presented in the current study indicate that the Dengue NS1 Panbio rapid test and the SD BIOLINE rapid test for malaria can play important roles in the early diagnosis of infections. Fluctuations in sensitivity and specificity can occur, however, depending upon the test manufacturer, study, and stage of the disease [43,44]. M-RT-PCR-ELISA identified 92% of the positive malaria rapid tests; this difference could be explained by the sensitivity and specificity of the rapid test. A study conducted in five malaria-endemic regions of Colombia found a Cohen’s kappa coefficient of 0.878 when comparing rapid test and PCR results; the sensitivity and specificity obtained for the rapid test were 94% and 95%, respectively [4]. Two additional positive results obtained using the rapid test were presumed to be false positives.

M-RT-PCR-ELISA also demonstrated the ability to identify parasitic and arboviral infections accurately. To make a reliable statement about the sensitivity and specificity of the method, however, requires analyses of a larger sample population. In order to determine the presence of previously undetected pathogens in a patient sample, the incorporation of additional pathogens into the m-RT-PCR-ELISA panel is currently under development.

The transmission of diseases like malaria and dengue is influenced by a variety of factors that include the environment, migration, deforestation, climate change, and sanitary conditions. The presence of arboviruses such as DENV, zika, and chikungunya has been reported mainly in urban areas, while parasitic infections such as malaria are found principally in rural areas. Cases of malaria have been reported in peri-urban areas, however, due to large migratory changes and the ability of vectors to reproduce in optimal climates below 2200 m above sea level [20]. Our study shows that although more cases of malaria are found in rural areas, changing environmental conditions favored the transmission of malaria in urban areas as well.

There was no significant difference between men and women, and the greatest number of cases occurred in patients over 15 years of age, which could be explained by social, behavioral, or occupational factors, where they could be more exposed to the vector’s bite.

An increased heart rate was observed among patients who were positive for dengue or malaria relative to individuals who were negative [45]. This increase might reflect a correlation with fever, which was frequently higher in disease-positive individuals. These results differ from those reported previously that showed a higher frequency of bradycardia among patients with dengue, although this sign may fluctuate [46]. Tachycardia, combined with other signs, i.e., respiratory distress, shock, severe bleeding, or organ involvement, could signal severe dengue. Taking tachycardia, along with these other signs, into account is important when collecting the patient’s medical history and considering DENV as the possible etiologic agent of AFS [47]. Clearly, the status of patients prior to enrollment in the current study was unknown. Therefore, heart rate could not be correlated with the evolution of disease.

In the case of the malaria-positive patients enrolled in this study, heart rate was lower than the rate determined in dengue-positive patients, a finding that differs from that reported previously in a study conducted in 2013 in French Guinea [48]. In the latter study, tachycardia >90 bpm was reported in a higher proportion of malaria-positive than -negative patients; tachycardia, however, is not directly related to case severity [49].

Patients positive for malaria had hemoglobin levels between 9.66 mg/dL and 14.54 mg/dL, similar to the 10.8–15 mg/d values determined for malaria-negative patients in the current study. Consequently, levels that fell within this range were not indicative of an ongoing malarial infection. It should be noted, however, that hemoglobin levels are an important indicator in cases of complicated malaria where values below 5 mg/dL (severe anemia) are suggestive of ongoing or severe disease [12].

Although it is recommended in clinical practice guidelines that anamnesis be performed as the main basis for diagnosis, the results of the current study demonstrate that the signs and symptoms exhibited by disease-positive and -negative patients, or by malaria-positive and dengue-positive patients did not differ significantly. Symptoms such as arthralgia, myalgia, fever, vomiting, and rash, however, were more frequent in patients positive for dengue. In a previous study, retro-orbital pain and rash were reported as the symptoms most characteristic of malaria [49]. These findings suggest that, despite a correct anamnesis, signs and symptoms alone are not conclusive. Rather, supporting diagnostic tests are required to diagnose and confirm a disease definitively [50].

The limitations in the current study include the timing of sample collection, as the number of intervening days between the onset of fever and sample collection is often unspecified. Given the maximum level of detection for some arboviruses (Zika, for example) requires sample collection within the first week of the first occurrence of symptoms, collection outside an optimal time interval could account for the failure to detect certain pathogens [51].

In the case of a single patient found positive for dengue by the rapid test in the present study, for example, dengue was not detected by m-RT-PCR-ELISA. Conceivably, the length of time between disease onset, sample collection, and sample analysis was an issue; viremia could already have fallen below the limits of m-RT-PCR-ELISA detection. Moreover, in cases involving coinfections with DENV serotypes, it is possible that viremia declines prematurely [25,52]. Additionally, it is important to consider that the sensitivity of multiplex PCR may be lower than single-primer PCR due to factors such as the method of specimen collection, viral load, and amplification conditions [53].

## 5. Conclusions

The presence of malaria and dengue was confirmed in both studied regions. m-RT-PCR-ELISA identified parasitic and arboviral pathogens in high concordance with the results of rapid testing.

## Figures and Tables

**Figure 1 tropicalmed-08-00466-f001:**
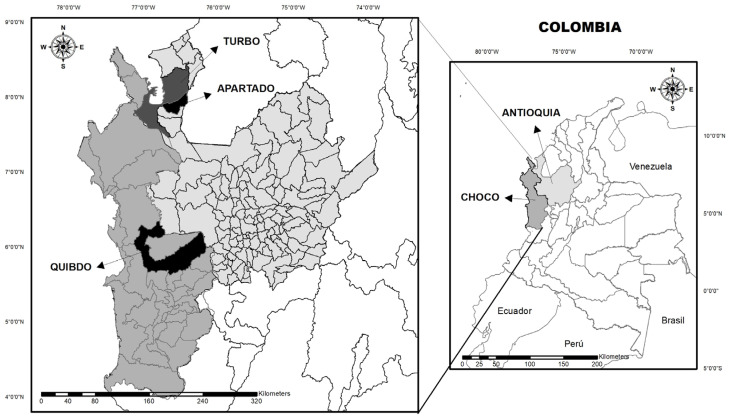
Geographical location of the study sites in Antioquia and Choco.

**Table 1 tropicalmed-08-00466-t001:** Baseline characteristics.

Sociodemographic Data	Total	
	n %	
Sex		
Male	83	53.5
Female	72	46.5
Age groups (years) ^a^		
<5	17	11.0
5 to ≤15	37	23.9
≥15	86	55.5
Study location		
Antioquia	81	52.3
Choco	74	47.7
Residence ^a^		
Urban area	101	65.2
Rural area	35	22.6
Village	8	5.2
History of malaria ^a^		
<1 month	1	0.6
1 to 2 months	3	1.9
3 to 6 months	8	5.2
7 to 12 months	3	1.9
>1 year	33	21.3
Unsure	2	1.3

Data are medians (IQR) derived from 155 patients, median age 24, range 8 to 42 years old. ^a^ Subdivision is missing data.

**Table 2 tropicalmed-08-00466-t002:** Signs and symptoms.

	Dengue-Positive Total (n = 16)		Malaria-Positive Total (n = 25)		Negative Total (n = 114)	
	n	%	n	%	n	%
Vital signs ^a^						
Heart rate (beats per min.)	108 (92–130)		98 (78–121)		86 (75–102)	
Respiratory rate (breaths per min.)	21 (19–22)		21 (18–24)		19 (16–22)	
Temperature °C	37.5 (37.1–38.5)		37.1 (36.6–37.9)		36.9 (36.3–37.8)	
SpO2 (%)	98 (98–99)		98 (97–99)		98 (97–98)	
Symptoms						
Headache	13	81.3	22	88.0	104	92.2
Chills	13	81.3	23	92.0	87	76.3
Sweats	11	68.8	19	76.0	82	71.9
Arthralgia	10	62.5	15	60.0	55	48.2
Nausea	8	50.0	13	52.0	54	47.4
Myalgia	9	56.3	13	52.0	52	45.6
Retro-orbital pain	6	37.5	12	48.0	55	48.2
Abdominal pain	7	43.8	12	48.0	36	31.6
Fever	8	50,0	9	36.0	35	30.7
Vomiting	7	43.8	7	28.0	25	21.9
Diarrhea	0	0.0	4	16.0	9	7.9
Rash	3	18.8	1	4.0	9	7.9
Hemoglobin (mg/dL) ^b^	12.5 ± 1.23		12.1 ± 2.44		12.9 ± 2.10	

Data are median (IQR). ^a^ Subdivision is missing data. ^b^ Data are the means ± SD.

**Table 3 tropicalmed-08-00466-t003:** Malaria rapid test and m-RT-PCR-ELISA results.

	Rapid Test ^a^			m-RT-PCR-ELISA		
	Result	n	%	Result	n	%
Antioquia	Negative	65	80.2	Negative	67	82.7
	*P. vivax*	9	11.2	*P. vivax*	8	9.9
	*P. falciparum*	7	8.6	*P. falciparum*	5 ^b^	6.2
Choco	Negative	65	87.8	Negative	65	87.8
	*P. falciparum*	5	6.8	*P. falciparum*	5	6.8
	*P. vivax*	4	5.4	*P. vivax*	4	5.4

^a^ Malaria SD BIOLINE. ^b^ One of the results could not be confirmed.

**Table 4 tropicalmed-08-00466-t004:** Dengue rapid test and m-RT-PCR-ELISA results.

	Rapid Test ^a^			m-RT-PCR-ELISA					
	Result	n	%	Result	n	%	Serotype	n	%
Antioquia	Negative	71	87.7	Negative	71	87.7			
	Positive	10	12.3	Positive	10	12.3	DENV2	1	10.0
							DENV3	4	40.0
							DENV1 & 4 ^b^	5	50.0
Choco	Negative	68	91.9	Negative	68	91.9			
	Positive	6	8.1	Positive	6	8.1	DENV1	2	33.3
							DENV1 & 4 ^b^	4	66.7

^a^ Dengue NS1 de Panbio^®^. ^b^ Coinfection.

## Data Availability

Data generated is not publicly available but can be requested from the corresponding author.

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
