# Peer review of "Multiplex Reverse Transcription Polymerase Chain Reaction Combined with a Microwell Hybridization Assay Screening for Arbovirus and Parasitic Infections in Febrile Patients Living in Endemic Regions of Colombia"

_tropicalmed, 2023, doi:10.3390/tropicalmed8100466_

Round 1

Reviewer 1 Report

Comments and Suggestions for Authors

The study applied a multiplex RT-PCR combined with a microwell hybridization assay (m-RT-PCR-ELISA) to screen 6 vector-borne pathogens in 155 patients in Antioquia and Choco, Colombia. Of 25 malaria patients identified by the rapid test, 23 were tested positive by the m-RT-PCR-ELISA (92%). In addition, 16 patients weredetected infected with dengue virus using the m-RT-PCR-ELISA and the Panbio® Dengue Early ELISA (Abbott, USA) rapid tests. The results obtained by the m-RT-PCR-ELISA were comparable to those obtained by commercial rapid tests.

The study intended to assess the feasibility of a m-RT-PCR-ELISA in diagnosis of vector-borne infections. However, several points needed to be clarified first.

1. The performance of m-RT-PCR-ELISA was only compared with commercial rapid tests. Usually, when assessing the sensitivity and specificity of a new test, it should be compared with the gold standards. 2. The advantages of m-RT-PCR-ELISA, e.g. portable, time-saving, less labor-demanding, cheaper, etc., were not specified. The readers cannot tell the reasons to use a new method. Especially when the authors only compared the results with commercial rapid tests, the advantagethe m-RT-PCR-ELISA had over the commercial rapid tests should be illustrated. 3. Although it has been mentioned in the abstract that no samples were positive for any of the other infectious agents tested (L22-23), the results should be described in the Results section of the context. 4. More details regarding the development of m-RT-PCR-ELISA should be described, e.g. the primers used, the determination of cut-off values, etc. 5. The test results were not analyzed against the demographic information although a questionnaire concerning sociodemographic data, signs and symptoms was completed at the time of admission.

Reviewer 2 Report

Comments and Suggestions for Authors

The manuscript ID: tropicalmed-2632856 describes a descriptive cross-sectional study conducted to evaluate the circulation and differential diagnosis of 6 pathogens in two departments of Colombia by multiplex reverse transcription polymerase chain reaction 19 combined with a microwell hybridization assay (m-RT-PCR-ELISA). The results displayed that M-RT-PCR-ELISA have the ability to identify parasitic and arboviral 245 pathogens in high concordance with the rapid test results. The paper is well written, methodologically fair, and well discussed. I have mostly just minor comments and I think it can be published after reviewing these little details.

1. Line 39 [4][5] ” → " [4,5] ”.

2. Line 54[8] [9] ”→ " [8,9] ”.

3. Please modify the reference format according to the magazine's requirements.

Comments on the Quality of English Language

the Quality of English Language is good

Reviewer 3 Report

Comments and Suggestions for Authors

In the opinion of this referee, the manuscript is clear, brief and concise. The focus is on the identification of Plasmodium, dengue virus, Zika virus, chikungunya virus, yellow fever virus and West Nile virus in hard-to-reach areas.

Understandably, the number of areas is limited. It would be preferable to increase the number of sampling sites but this is understandably difficult and justified. The 155 patients is a good number but if the number of samples in each district could be homogenised, i.e. a representative number based on the population of the area, if possible, it would be advisable or, if not, justified. It is not necessary for the manuscript but if it could be done, it would clarify this fact.

Round 2

Reviewer 1 Report

Comments and Suggestions for Authors

The revised version is improved and is suitable for publication.